# Associations between Age, Body Composition, Balance, and Other Physical Fitness Parameters in Youth Soccer

**Cíntia França** [1,2], **Francisco Martins** [1,2], **Adilson Marques** [3,4,*], **Marcelo de Maio Nascimento** [5], **Andreas Ihle** [6,7,8], **Krzysztof Przednowek** [9] **and Élvio Rúbio Gouveia** [1,2,7]

1 Department of Physical Education and Sport, University of Madeira, 9020-105 Funchal, Portugal
2 Interactive Technologies Institute, LARSYS, 9020-105 Funchal, Portugal
3 CIPER, Faculty of Human Kinetics, University of Lisbon, 1495-751 Lisbon, Portugal
4 ISAMB, Faculty of Medicine, University of Lisbon, 1649-020 Lisbon, Portugal
5 Department of Physical Education, Federal University of Vale do São Francisco, Petrolina 56304-917, Brazil
6 Department of Psychology, University of Geneva, 1205 Geneva, Switzerland
7 Center for the Interdisciplinary Study of Gerontology and Vulnerability, University of Geneva, 1205 Geneva, Switzerland
8 Swiss National Centre of Competence in Research LIVES—Overcoming Vulnerability: Life Course Perspectives, 1015 Lausanne, Switzerland
9 Institute of Physical Culture Sciences, Medical College, University of Rzeszów, 35-959 Rzeszów, Poland
* Correspondence: amarques@fmh.ulisboa.pt

**Abstract:** In sports, balance ability has been related to game performance and injury prevention. This study's aims were twofold: (1) to analyze the balance performance of adolescent soccer players from different age groups; and (2) to examine the relationship between players' age, body composition, balance, and other physical fitness parameters, such as strength and flexibility. In this study, 112 players from the under 15 (U15), under 16 (U16), and under 17 (U17) age groups participated. A one-way analysis of the variance was conducted to investigate differences between groups. Pearson correlations and hierarchical multiple regression were used to explore the relationship between variables. Regarding balance, the older group performed significantly worse in the stability indexes for both legs but significantly better in dynamic balance tests. Height correlated significantly and negatively with balance indicators. Both jumping tasks showed medium to large correlations with the sway indexes ($-0.23 > r < -0.51$). Although not significantly, body fat negatively affected balance, underlining the importance of monitoring body composition for players' development. Overall, no substantial relationship was found between static and dynamic balance variables, and therefore, it is crucial to include both as complementary measures while evaluating youngsters' postural balance.

**Keywords:** strength; age; posture; youngsters; flexibility; vertical jumping

## 1. Introduction

Balance is defined as the state of an object when the resultant load actions, such as forces or movements acting upon it, are zero [1]. The ability to maintain balance in a static position is related to the position of the body's center of mass (CoM) and its area of the base of support. If the line of gravity falls within the body's base of support, the individual is balanced [2]. Therefore, balance emerges from the interaction between the individual, the environment, and the task [3]. Functional tasks may require steady, reactive, or proactive balance control, while environmental constraints, such as the type of support surface or cognitive demands, influence balance control. On the other hand, individual variations in motor, sensory and cognitive abilities contribute to generating the motor output that allows the maintenance of a controlled posture [4,5].

In sports, balance ability has been associated with performance and injury prevention [6]. In soccer, past studies have reported differences in dynamic balance performance

according to the competition level, with the more proficient players displaying greater balance ability than their lower-division peers [6–8]. Moreover, balance training has been documented to enhance athletic performance and contribute to the prevention and rehabilitation of injury [9–11]. However, most previous investigations on balance have privileged adult or elite players, and research among youngsters is still lacking.

Sports are the leading cause of injury in youth, which may compromise future physical activity participation and adversely affect future health [12]. In addition, enhancing youth players' physical development is crucial to improving their long-term athletic performance. Although several works have been promoted on this topic, most of the investigations were focused on studying youngsters' body composition [13], strength [14,15], speed and agility [16,17], and aerobic and anaerobic performance [18,19]. To the best of our knowledge, among physical fitness components, which comprise health-related (body composition, cardiorespiratory endurance, flexibility, muscular strength and endurance) and skill-related factors (balance, agility, and coordination) [20], balance is one of the less studied areas in youth sports.

Soccer involves performing several actions using a unipedal stance, such as kicking, passing, and dribbling, often demanding the need to control body sway [21]. Most of the balance topic research has focused on the effects of balance training programs on physical performance [22,23] and injury prevention [24]. For example, one investigation examined the effects of balance and plyometric training among 24 youth soccer players aged $12.7 \pm 0.3$ years, and it reported the benefits of both interventions in jumping and sprinting capacities [22]. On the other hand, one study found the benefits of balance training in the performance of soccer-specific skills, such as kicking [25].

Although the effects of balance training to improve health and skill-related components, such as sprinting, jumping, and sports-specific skills, among young athletes is well established [23], details are still needed concerning the interrelationship between chronological age (CA) balance, body composition and strength performance in youth soccer. Age has been proven to influence physical fitness performance, particularly in youth, due to its strong relationship with individuals' growth and experience levels [26–28]. The literature has also described that balance strategies during gait are task-specific and vary according to age [29]. A past investigation in youth soccer concluded that older players (U19) showed significantly better performance in the Lower Quarter Y Balance Test compared with their younger peers (U13) [30]. However, in the overall soccer context, research on balance topic has focused on the effectiveness of training programs to enhance balance ability or physical performance.

Therefore, the aims of the present study were twofold: (1) to analyze the balance performance of adolescent male soccer players from different age groups; and (2) to examine the relationship between players' age, body composition, balance, and other physical fitness parameters, such as strength and flexibility. It was hypothesized that: (1) older players should present better balance performance than younger players; and (2) age, body composition, strength, and flexibility variables would present a strong relationship with balance performance.

## 2. Materials and Methods

One hundred and twelve male soccer players from the under 15 (U15), under 16 (U16), and under 17 (U17) age groups participated in this study. All participants were competing at the regional level in Portugal. All the assessments were performed in a physical performance laboratory 5 min apart between protocols. The protocols were applied by trained staff from the research team. The procedures applied in this study were approved by the Ethics Committee of the Faculty of Human Kinetics, CEIFMH N°34/2021, and followed the Declaration of Helsinki. Participation in this study was voluntary, and informed consent was obtained from the youngsters' legal guardians.

### 2.1. Body Composition

Height was measured to the nearest 0.01 cm using a stadiometer (SECA 213, Hamburg, Germany), and body composition was measured using a hand-to-foot bioelectrical impedance analysis (InBody 770, Cerritos, CA, USA). The measurement occurred in the early morning for five consecutive days. A group of 25–30 players was evaluated each day, and the mean time between the first and the last evaluation was about 30 min. At the assessment, participants were fasting and wearing only their underwear. On the platform, participants were barefoot with their feet placed on the defined spots and standing with their arms nearly 45° from their trunk. Body mass, body fat percentage (BF%), and fat-free mass (FFM) were used for analysis.

### 2.2. Handgrip

The handgrip protocol included three alternated trials for each arm using a hand dynamometer (Jamar Plus+, Chicago, IL, USA) [31] The rest interval between trials was 60 s. Participants were asked to hold the dynamometer in one hand, laterally to their trunk with the elbow at a 90° position while standing. Then, participants were asked to squeeze as hard as possible for about two seconds. The best score of the three trials was retained for analysis.

### 2.3. Sit-Ups

The sit-ups protocol consisted of performing the maximal number of repetitions within 30 s [32]. The participants started in a sitting position, with the torso vertical, hands behind their neck, and knees bent at a 90° position with the feet placed on the floor. Participants were instructed to stretch out on their back with shoulders touching the floor, straighten up to the sitting position, bring elbows forward in contact with their knees, and/or pass them through the knees. One member of the research team performed the counting. One repetition was considered at the moment when the elbows touched or passed the knees. The absence of counting meant that the repetition had not been correctly performed. The total number of repetitions made was used for analysis.

### 2.4. Vertical Jumping

Lower-body explosive strength was evaluated using the countermovement jump (CMJ) and the squat jump (SJ) [33]. Both protocols included four data collection trials performed 30 s apart. The maximum height attained during the jumps was recorded using the Optojump Next (Microgate, Bolzano, Italy) system of analysis and measurement, and the best score remained for analysis. Before data collection, the participants performed three experimental trials to guarantee correct execution. The CMJ protocol began in a standing position, with feet placed hip-width to shoulder-width apart. Then, participants executed a countermovement to a depth position close to 90° of knee flexion, which was followed by a maximal-effort vertical jump. During the executions, the hands remained on the hips for the entire movement to avoid the influence of arm swing. If excessive knee flexion was observed or if the hands were removed from the hips, the trial was repeated. For the SJ, participants began squatting at nearly 90° of knee flexion depth. From this position, participants were asked to jump to maximum height. The trial was repeated if a dipping movement of the hips was evident before the jump. After each jump, the starting position was reset.

### 2.5. Flexibility

Flexibility measurements were evaluated using two trials of the sit and reach tests [32]. A trunk flexibility box (32.4 cm high and 53.3 cm long) with a 23 cm heel line mark was used. The protocol was initiated with participants sitting barefoot in front of the box for the unilateral evaluation, with the knee fully extended and the heel placed against the box. Afterwards, participants placed their hands on each other and slowly bent forward along the measuring scale. Participants were asked to achieve the maximum forward position

and to hold onto that position for about 3 s. The first procedure was performed with the right leg and then the left. The same protocol was applied for the bilateral measurement, placing both heels against the box. The best score attained was retained for analysis.

*2.6. Balance*

Balance was assessed using the Biodex Balance System SD (Biodex, Shirley, NY, USA). Before each testing session, the equipment was adjusted to the participant's height. Participants were allowed to practice with the protocols through a single training session to guarantee the protocols' understanding and minimize learning effects later on during the testing phase. The rest interval between testing sessions was set at 60 s.

For bilateral comparison, the protocol was performed in a unilateral stance with participants barefoot. During the assessment, the overall stability index (OSI), anteroposterior stability index (APSI), and mediolateral stability index (MLSI) were measured under four levels of platform stability for 20 s. Level 4 was the most stable, and level 1 was the most unstable. The scores for the indexes show the level of deviation from the horizontal position; therefore, lower scores indicate better balance [34].

Then, participants were submitted to the modified clinical test of sensory interaction and balance (mCTSIB) under four different conditions in the following order: eyes open hard surface (EOHS), eyes closed hard surface (ECHS), eyes open soft surface (EOSS), and eyes closed soft surface (ECSS). For testing, participants were barefoot in an upright position, arms placed laterally to the body, and feet set shoulder-width apart. Each mCTSIB condition was conducted once for 30 s. The score of sway index (SI), which corresponds to the deviation of the center of pressure position from the mean center of pressure position during the entire trial, was used for analysis. Higher SI values indicate a diminished ability to maintain balance during testing [35].

*2.7. Statistics*

Descriptive statistics were presented as means $\pm$ standard deviation. All data were checked for normality using the Shapiro–Wilk test. A one-way analysis of the variance (ANOVA) and post hoc comparisons using the Bonferroni adjustment was conducted to investigate differences in age and physical fitness components. The relationship between body composition, strength, and balance performance was explored using Pearson product-moment correlation coefficient. Finally, hierarchical multiple regression analyses were used to investigate the amount of variance in dynamic balance performance explained by strength indicators (entered in step 3) after controlling for CA (entered in step 1) and body composition (entered in step 2). All analyses were conducted using IBM SPSS Statistics software 28.0 (SPSS Inc., Chicago, IL, USA). The significance level was set at $p \leq 0.05$.

## 3. Results

Table 1 summarizes sample descriptive statistics regarding age and physical fitness components. The results of one-way ANOVA between groups suggest significant differences in body composition, strength, and balance tests. The U15 group presented significantly lower body mass (F = 27.264, $p \leq 0.01$), greater BF% (F = 5.848, $p \leq 0.01$), and lower FFM (F = 55.359, $p \leq 0.01$) than their older peers. Concerning FFM, the U16 group also showed a substantially lower mean value than the U17 group. In strength tests, the U17 group significantly outperformed their younger counterparts, particularly in the SJ (F = 44.405, $p \leq 0.01$). Overall, the groups did not differ significantly in flexibility.

In contrast, the balance indicators observed substantial differences between the U15 and the U17 groups. The older group performed significantly worse in both legs' stability indexes (OSI, APSI, and LMSI). However, regarding the sway indexes, the older group achieved significantly better results than the youngest in all four conditions.

Tables 2 and 3 present the Pearson product-moment correlation coefficient results used to explore the relationship between age, body composition, strength, flexibility, and balance tests. CA showed a positive and significant correlation with stability indexes, particularly

with OSI and APSI at the right side (r = 0.50; $p \leq 0.01$). In contrast, CA displayed a significant positive relationship with sway indexes, particularly on the EOSS condition (r = −0.50; $p \leq 0.01$), while height correlated significantly and negatively with balance variables. Body mass and FFM also presented significant correlations with balance, with a negative relationship observed with the stability indexes and a positive association found with the sway indexes. As expected, there was a large correlation between the stability indexes and also a large correlation among the sway indexes. However, no relationships were observed between stability and sway indexes.

**Table 1.** Descriptive statistics for age, body composition, flexibility, strength, and balance tests among youth soccer players (n = 112).

| Variable | U15 (n = 53) | U16 (n = 20) | U17 (n = 39) | ANOVA | | |
|---|---|---|---|---|---|---|
| | Mean ± SD | Mean ± SD | Mean ± SD | F | p | Post Hoc Comparisons |
| CA (years) | 13.8 ± 0.1 | 15.6 ± 0.1 | 16.8 ± 0.2 | 272.204 | ≤0.01 | U15 < U16 and U17; U16 < U17 |
| Height (cm) | 164.4 ± 1.1 | 173.8 ± 2.0 | 175.1 ± 2.2 | 17.335 | ≤0.01 | U15 < U16 and U17 |
| Body mass (kg) | 54.7 ± 1.4 | 63.1 ± 2.2 | 66.3 ± 2.3 | 27.264 | ≤0.01 | U15 < U16 and U17 |
| BF (%) | 14.6 ± 1.0 | 11.7 ± 1.0 | 9.8 ± 0.8 | 5.848 | ≤0.01 | U15 > U17 |
| FFM (kg) | 46.3 ± 0.9 | 55.5 ± 1.7 | 59.7 ± 1.9 | 55.359 | ≤0.01 | U15 < U16 and U17, U16 < U17 |
| Handgrip (kg) | 28.3 ± 0.8 | 37.1 ± 1.4 | 38.9 ± 1.7 | 39.093 | ≤0.01 | U15 < U16 and U17 |
| Sit-ups (n) | 25.5 ± 0.4 | 24.9 ± 0.8 | 27.9 ± 1.1 | 6.873 | ≤0.01 | U15 < U17, U16 < U17 |
| CMJ height (cm) | 26.6 ± 0.6 | 32.9 ± 0.8 | 34.4 ± 0.8 | 43.170 | ≤0.01 | U15 < U16 and U17 |
| SJ height (cm) | 26.2 ± 0.6 | 31.1 ± 0.8 | 32.9 ± 0.8 | 44.405 | ≤0.01 | U15 < U16 and U17, U16 < U17 |
| Flexibility unilateral (cm) | 30.2 ± 0.8 | 33.6 ± 1.7 | 31.3 ± 1.1 | 2.076 | 0.13 | |
| Flexibility bilateral (cm) | 29.5 ± 0.9 | 33.3 ± 1.7 | 32.0 ± 1.2 | 2.566 | 0.08 | |
| OSI left (°) | 1.67 ± 0.12 | 1.88 ± 0.31 | 3.25 ± 0.36 | 11.819 | ≤0.01 | U15 < U16; U16 < U17 |
| APSI left (°) | 1.03 ± 0.09 | 1.20 ± 0.30 | 2.32 ± 0.36 | 8.766 | ≤0.01 | U15 < U16; U16 < U17 |
| LMSI left (°) | 1.09 ± 0.10 | 1.16 ± 0.16 | 1.89 ± 0.19 | 9.903 | ≤0.01 | U15 < U16; U16 < U17 |
| OSI right (°) | 1.23 ± 0.06 | 1.17 ± 0.19 | 2.97 ± 0.40 | 16.417 | ≤0.01 | U15 < U16; U16 < U17 |
| APSI right (°) | 0.81 ± 0.05 | 0.88 ± 0.18 | 2.49 ± 0.41 | 14.291 | ≤0.01 | U15 < U16; U16 < U17 |
| LMSI right (°) | 0.72 ± 0.05 | 0.59 ± 0.06 | 1.17 ± 0.14 | 9.496 | ≤0.01 | U15 < U16; U16 < U17 |
| SI EOHS | 0.92 ± 0.09 | 0.70 ± 0.07 | 0.68 ± 0.09 | 4.910 | ≤0.01 | U15 > U17 |
| SI ECHS | 1.28 ± 0.09 | 1.14 ± 0.12 | 0.91 ± 0.08 | 8.345 | ≤0.01 | U15 > U17 |
| SI EOSS | 1.09 ± 0.06 | 0.85 ± 0.05 | 0.74 ± 0.04 | 11.942 | ≤0.01 | U15 > U16 and U17 |
| SI ECSS | 2.50 ± 0.08 | 2.36 ± 0.10 | 2.26 ± 0.19 | 3.191 | 0.05 | U15 > U17 |

SD (standard deviation); BF (body fat); FFM (fat free mass); CMJ (countermovement jump); SJ (squat jump); OSI (overall stability index); APSI (anteroposterior stability index); LMSI (lateromedial stability index); SI EOHS (sway index eyes open hard surface); SI ECHS (sway index eyes close hard surface); SI EOSS (sway index eyes open soft surface); SI ECSS (sway index eyes close soft surface).

Regarding strength, a large and positive relationship was seen between the handgrip and vertical jumping (CMJ: r = 0.53; $p \leq 0.01$, SJ: r = 0.52; $p \leq 0.01$). In contrast, the handgrip was significantly and negatively associated with the sway indexes, with the highest correlation corresponding to the EOSS condition (r = −0.52; $p \leq 0.01$). Both jumping tasks showed medium to large correlations with the sway indexes. In contrast, among the flexibility tests, only the unilateral test showed substantial correlations with some stability indexes (OSI left, LMSI left, and LMSI right).

Finally, the results of hierarchical multiple regression conducted to investigate the effects of strength indicators (entered in step 3) on dynamic balance performance after controlling for CA (entered in step 1) and body composition (entered in step 2) are shown in Table 4. CA alone was a significant predictor of dynamic balance performance in all four conditions. However, after entering body composition variables, the effects of CA tended to disappear. Overall, CA and body composition explained between 9% (ECSS condition) and 34% (EOSS condition) of the variance observed in dynamic balance tests. The introduction of strength indicators only explained an additional small amount observed in performance. The sit-ups were the only tests remaining as significant after controlling for CA and body composition, particularly in the ECSS condition.

**Table 2.** Significant correlation coefficients between age, body composition, and balance of adolescent male soccer players (n = 112).

| Variable | 1. | 2. | 3. | 4. | 5. | 6. | 7. | 8. | 9. | 10. | 11. | 12. | 13. | 14. | 15. |
|---|---|---|---|---|---|---|---|---|---|---|---|---|---|---|---|
| 1. CA | - | 0.61 ** | 0.64 ** | −0.22 * | 0.74 ** | 0.44 ** | 0.40 ** | 0.38 ** | 0.50 ** | 0.50 ** | 0.32 ** | −0.35 ** | −0.44 ** | −0.50 ** | −0.26 ** |
| 2. Height | | - | 0.75 ** | −0.24 * | 0.89 ** | −0.29 ** | −0.26 * | −0.25 * | −0.33 ** | −0.23 * | −0.31 ** | −0.25 * | −0.23 * | −0.51 ** | |
| 3. Body mass | | | - | 0.19 * | 0.90 ** | 0.28 ** | 0.26 ** | 0.24 * | 0.29 ** | 0.27 ** | 0.27 ** | −0.27 ** | −0.25 ** | −0.48 ** | −0.19 * |
| 4. BF% | | | | - | −0.24 * | | | | | | | 0.22 * | 0.36 ** | 0.23 * | |
| 5. FFM | | | | | - | 0.26 ** | 0.25 ** | 0.22 * | 0.32 ** | 0.31 ** | 0.26 ** | −0.34 ** | −0.38 ** | −0.55 ** | −0.25 ** |
| 6. OSI left | | | | | | - | 0.94 ** | 0.78 ** | 0.80 ** | 0.77 ** | 0.60 ** | | | | |
| 7. APSI left | | | | | | | - | 0.54 ** | 0.80 ** | 0.80 ** | 0.51 ** | | | | |
| 8. LMSI left | | | | | | | | - | 0.56 ** | 0.49 ** | 0.61 ** | | | | |
| 9. OSI right | | | | | | | | | - | 0.98 ** | 0.60 ** | | | | |
| 10. APSI right | | | | | | | | | | - | 0.43 ** | | | | |
| 11. LMSI right | | | | | | | | | | | - | | | | |
| 12. SI EOHS | | | | | | | | | | | | - | 0.68 ** | 0.68 ** | 0.39 ** |
| 13. SI ECHS | | | | | | | | | | | | | - | 0.63 ** | 0.43 ** |
| 14. SI EOSS | | | | | | | | | | | | | | - | 0.33 ** |
| 15. SI ECSS | | | | | | | | | | | | | | | - |

CA (chronological age); BF (body fat); FFM (fat free mass); OSI (overall stability index); APSI (anteroposterior stability index); LMSI (lateromedial stability index); SI EOHS (sway index eyes open hard surface); SI ECHS (sway index eyes close hard surface); SI EOSS (sway index eyes open soft surface); SI ECSS (sway index eyes close soft surface); * $p \leq 0.05$; ** $p \leq 0.01$.

**Table 3.** Correlation coefficients between strength, flexibility, and balance of adolescent male soccer players (n = 112).

| Variable | 1. | 2. | 3. | 4. | 5. | 6. | 7. | 8. | 9. | 10. | 11. | 12. | 13. | 14. | 15. | 16. |
|---|---|---|---|---|---|---|---|---|---|---|---|---|---|---|---|---|
| 1. Handgrip | - | 0.19 * | 0.53 ** | 0.52 ** | | | | | | 0.19* | | | −0.33 ** | −0.34 ** | −0.52 ** | −0.24 * |
| 2. Sit-ups | | - | 0.28 ** | 0.35 ** | | | | | | | | | −0.19 * | −0.23 * | | −0.24 * |
| 3. CMJ height | | | - | 0.95 ** | | | | 0.20 * | | 0.19 * | 0.20 * | | −0.32 ** | −0.39 ** | −0.43 ** | −0.23 * |
| 4. SJ height | | | | - | | | 0.25 ** | 0.26 ** | | 0.31 ** | 0.32 ** | | −0.34 ** | −0.42 ** | −0.41 ** | −0.23 * |
| 5. Flexibility unilateral | | | | | - | 0.87 ** | 0.20 * | | 0.26 ** | | | 0.25 ** | | | | |
| 6. Flexibility bilateral | | | | | | - | | | | | | | | | | |
| 7. OSI left | | | | | | | - | 0.95 ** | 0.78 ** | 0.80 ** | 0.77 ** | 0.60 ** | | | | |
| 8. APSI left | | | | | | | | - | 0.54 ** | 0.80 ** | 0.80 ** | 0.51 ** | | | | |
| 9. LMSI left | | | | | | | | | - | 0.56 ** | 0.49 ** | 0.61 | | | | |
| 10. OSI right | | | | | | | | | | - | 0.98 ** | 0.60 ** | | | | |

**Table 3.** *Cont.*

| Variable | 1. | 2. | 3. | 4. | 5. | 6. | 7. | 8. | 9. | 10. | 11. | 12. | 13. | 14. | 15. | 16. |
|---|---|---|---|---|---|---|---|---|---|---|---|---|---|---|---|---|
| 11. APSI right | | | | | | | | | | | - | 0.43 ** | | | | |
| 12. LMSI right | | | | | | | | | | | | - | | | | |
| 13. SI EOHS | | | | | | | | | | | | | - | 0.68 ** | 0.68 ** | 0.39 ** |
| 14. SI ECHS | | | | | | | | | | | | | | - | 0.63 ** | 0.43 ** |
| 15. SI EOSS | | | | | | | | | | | | | | | - | 0.33 ** |
| 16. SI ECSS | | | | | | | | | | | | | | | | - |

CMJ (countermovement jump); SJ (squat jump); OSI (overall stability index); APSI (anteroposterior stability index); LMSI (lateromedial stability index); SI EOHS (sway index eyes open hard surface); SI ECHS (sway index eyes close hard surface); SI EOSS (sway index eyes open soft surface); SI ECSS (sway index eyes close soft surface); * $p \le 0.05$; ** $p \le 0.01$.

**Table 4.** Summary of hierarchical regression analysis with strength indicators predicting dynamic balance after controlling for CA and body composition.

| Variable | SI EOHS | | | SI ECHS | | | SI EOSS | | | SI ECSS | | |
|---|---|---|---|---|---|---|---|---|---|---|---|---|
| | Model I | Model II | Model III | Model I | Model II | Model III | Model I | Model II | Model III | Model I | Model II | Model III |
| | β | β | β | β | β | β | β | β | β | β | β | β |
| CA | −0.34 ** | −0.15 | −0.07 | −0.42 ** | −0.26 * | −0.22 | −0.50 ** | −0.17 | −0.17 | −0.25 ** | −0.14 | −0.13 |
| Body mass | | −0.22 | −0.16 | | −0.13 | −0.15 | | 0.42 ** | 0.28 | | −0.12 | −0.07 |
| BF% | | 0.23 * | 0.18 | | 0.33 ** | 0.34 ** | | 0.28 ** | 0.20 | | 0.17 | 0.16 |
| Handgrip | | | −0.09 | | | 0.02 | | | −0.14 | | | −0.05 |
| Sit-ups | | | −0.08 | | | −0.12 | | | −0.02 | | | −0.21 * |
| SJ height | | | 0.11 | | | 0.12 | | | −0.30 | | | −0.09 |
| CMJ height | | | −0.18 | | | −0.12 | | | 0.27 | | | 0.16 |
| $R^2$ | 0.12 | 0.16 | 0.17 | 0.17 | 0.26 | 0.28 | 0.25 | 0.34 | 0.36 | 0.07 | 0.09 | 0.12 |
| *F* for change in $R^2$ | 13.755 ** | 6.514 ** | 2.984 ** | 22.462 ** | 12.190 ** | 5.452 ** | 34.163 ** | 17.851 ** | 7.892 ** | 7.241 ** | 3.253 * | 1.991 |

Model I: CA; Model II: CA, body mass and BF%; Model III: CA, body mass, BF%, Handgrip, Sit-ups, SJ height and CMJ height. CA (chronological age); BF% (body fat percentage); SJ (squat jump); CMJ (countermovement jump). * $p \le 0.05$; ** $p \le 0.01$.

## 4. Discussion

This study aimed to analyze the balance performance of adolescent male soccer players from different age groups and to examine the relationship between age, body composition, balance, and other selected physical fitness components. Our results showed that younger players (U15) performed significantly better in the BLC tests than the older players (U17), which is not in line with our first hypothesis. In contrast, the U17 players considerably outperformed their younger peers in the mCTSIB tests. As expected, age showed a large relationship with balance performance, which was followed by height, body mass, and vertical jumping. After controlling for CA and body composition, the strength indicators did not remain as significant predictors of dynamic balance, except for sit-ups in the ECSS condition. Flexibility (unilateral) only showed a significant relationship with three stability indexes.

The comparison between age groups revealed that the U17 group was substantially taller and heavier and presented lower BF% and greater FFM than their younger peers. In addition, the U17 group performed significantly better in static strength (handgrip), muscular strength and endurance (sit-ups), and lower-body explosive strength (CMJ and SJ), compared to the U16 and U15 groups. In youth sports, the influence of age on athletic performance has been demonstrated by increased body size and superior levels of strength and power, which was mainly due to biological maturation [36]. Although biological maturation varies in timing and tempo [37,38], the literature has suggested an average age at peak height velocity for samples of European boys ranging between 13.8 and 14.2 years [39]. This age range is covered in our sample. Therefore, differences in body composition and strength could reflect the influence of players' maturity status.

In contrast, no substantial differences were observed between groups concerning flexibility tests. The U16 players showed better performance levels both in the unilateral and the bilateral testing. According to past studies, flexibility tends to decrease over age due to the diminution of the range of motion [39–41], which underlines the need to consider systematic training specifically focused on developing this capacity [42].

Regarding balance, our findings suggest a significantly greater static balance performance by younger players and a considerably greater dynamic balance performance by older players. Static balance is related to maintaining a base of support with minimal movement. In contrast, dynamic balance concerns the ability to perform a task while maintaining a stable position [43]. Several factors, among others, may influence balance ability, such as age, height, body mass, and sport participation level [44]. Previous research among 130 youth male soccer players aged between 10 and 18 years reported similar results to the ones found in our study. Although the methods used to assess balance differed from ours, younger players significantly outperformed their peers in static balance, while older players showed better dynamic balance [45]. Indeed, the literature has mentioned that postural sway has been shown to increase with age, while timed unipedal balance tends to decrease with age [44], which is corroborated by our results.

A better understanding of these results may be gained by interpreting the correlation analyses conducted in this study. First, it seems important to underline the absence of no statistically significant correlation between BLC and mCTSIB tests, with the respective values ranging from −0.01 to −0.17. These results are in line with past investigations in different populations [46,47]. Although static and dynamic balance control involves the same neural structures (i.e., cerebral cortex, cerebellum, spinal cord), their complementary contributions seem to be different for the two testing conditions (static vs. dynamic), accounting for the non-significant correlations detected [46]. According to our analyses, age presented the highest number of relationships with balance tests, followed by height, body mass, and FFM. The association between CA and balance was stronger regarding BLC, suggesting that older players should present superior scores in BLC tests, which indicates lower performance levels. In contrast, CA was significantly and negatively related to mCTSIB, suggesting that older players should attain lower scores in mCTSIB tests, which corresponds to superior performance levels.

Meanwhile, body mass and FFM demonstrated substantial relationships with balance, particularly in dynamic conditions. On the other hand, the association between height and balance indicators was significantly negative, which suggests that shorter individuals may have advantages regarding postural balance. No previous studies on the relationship between body mass, FFM, height, and balance were found. However, the literature has described that a higher body mass index demands more displacements to maintain postural balance [48].

The hierarchical regression analyses showed BF% as a significant negative predictor of dynamic balance performance, even after controlling for CA. Although the effects of BF% were not substantial after introducing strength variables in the model, it is still recommended to monitor players' body composition, mainly to avoid the detrimental influence of BF% on sports performance [26,39,49].

Regarding strength, vertical jumping tasks showed the highest number of significant correlations with balance. However, after controlling for CA and body composition, the strength indicators were not significant predictors of dynamic balance except for the sit-ups in the ECSS condition. Past research described high levels of activity of trunk muscles when necessary to stabilize the trunk over a base of support [50,51], which emphasizes the role of the core in balance tasks.

On the other hand, in our analyses, the whole model (CA, body composition, and strength) could explain between 12 and 36% of the variance observed in the dynamic balance performance. Although CA alone was a substantial predictor of balance, this effect tended to disappear while body composition and strength were introduced in the model. The linear relationships between strength and balance tests suggest the need to include strength contents during the soccer training process to improve balance and vice versa. According to the literature, the enhanced balance has been reported to improve strength [52], which could positively influence the rate of force development, including dynamic activities such as the SJ and the CMJ [52,53]. Introducing a 4-week balance-training program into physical education classes has significantly improved postural control, jumping height, and the rate of force development [52]. In another study among soccer players, the authors described medium to large associations between balance, back extensor strength, and jumping ability [45]. However, the literature has mentioned that the significant relationships between dynamic balance and lower-body explosive strength depend on each other [54]. Therefore, including balance content in the players' training process may be helpful to enhance not only postural control but also strength. In addition, previous research has recommended using strength exercises to improve balance [54], providing the overall players' development.

This study is based on cross-sectional data and did not assess the players' maturity status, representing its limitations. Indeed, individual development is age-related, particularly during adolescence. Therefore, longitudinal data would be far more informative regarding youngsters' profiles. However, to the best of our knowledge, this is one of the first and novel investigations focused on evaluating balance indicators among youth soccer players and their relationship with other physical fitness variables. No substantial relationship was found between static and dynamic balance variables. Thus, it is crucial to include both as complementary measures while performing postural balance assessments among youth soccer players.

On the other hand, CA showed the highest number of correlations with balance indicators. CA alone was a substantial predictor of dynamic balance. However, this effect disappeared after introducing the model's body composition and strength variables. Our results, together with the literature, which has described the strong and positive association between balance, motor skills acquisition, motor performance enhancement, and injury prevention [55–57], emphasize the need to promote training strategies to improve balance among youngsters with fewer balance abilities. This would benefit soccer game performance, particularly in actions such as dribbling, passing, and gaining positions between opponents. Due to the interrelationship between strength and balance, strength

contents could be used to improve balance. In addition, body composition should be monitored mainly to avoid the detrimental effects of BF% on balance performance.

## 5. Conclusions

Our study indicates differences in the balance assessment among youth soccer players according to age. Younger players presented better static balance performance, while older players showed superior dynamic balance ability. Age was largely correlated with balance, which was followed by height, body mass, and vertical jumping. However, after controlling for CA and body composition, the strength indicators did not remain significant predictors of dynamic balance, except for sit-ups in the ECSS condition. At the same time, the linear relationships between strength and balance tests suggest the need to include strength contents during the soccer training process to improve balance and vice versa. Although not significant, BF% showed a detrimental effect on balance, which underlines the importance of monitoring body composition during the players' development. Overall, no substantial relationship was found between static and dynamic balance variables. This is a cross-sectional study, and players' maturity status was not assessed, representing its limitations. Longitudinal data and considering youngsters' maturity status would be far more informative and should be considered in future works. Our results suggest including both types of balance assessment (static and dynamic) as complementary measures while evaluating youngsters' postural balance.

**Author Contributions:** Conceptualization, C.F., F.M. and É.R.G.; methodology, C.F., F.M., M.d.M.N., K.P. and É.R.G.; software, C.F. and F.M.; validation, A.M., A.I., É.R.G. and É.R.G.; formal analysis, C.F., F.M., M.d.M.N. and É.R.G.; investigation, C.F. and F.M.; resources, A.M., A.I. and É.R.G.; data curation, C.F., F.M. and É.R.G.; writing—original draft preparation, C.F. and F.M.; writing—review and editing, C.F., F.M., M.d.M.N., É.R.G., K.P., A.M. and A.I.; visualization, A.M., A.I., K.P. and É.R.G.; supervision, É.R.G.; project administration, É.R.G.; funding acquisition, A.M., A.I. and É.R.G. All authors have read and agreed to the published version of the manuscript.

**Funding:** This work was supported by the Swiss National Centre of Competence in Research LIVES, "Overcoming Vulnerability: Life Course Perspectives," granted by the Swiss National Science Foundation (Grant Number: 51NF40-185901). A.I. acknowledges support from the Swiss National Science Foundation (Grant Number: 10001C_189407). C.F., F.M. and É.R.G. acknowledge support from LARSyS—Portuguese national funding agency for science, research, and technology (FCT) pluriannual funding 2020–2023 (Reference: UIDB/50009/2020). This study is framed in the Marítimo Training Lab Project. The project received funding under application no. M1420-01-0247-FEDER-000033 in the System of Incentives for the Production of Scientific and Technological Knowledge in the Autonomous Region of Madeira—PROCiência 2020.

**Institutional Review Board Statement:** This study was conducted according to the guidelines of the Declaration of Helsinki and approved by the Ethics Committee of the Faculty of Human Kinetics, (CEIFMH N° 34/2021), and it followed the ethical standards of the Declaration of Helsinki for Medical Research in Humans (2013) and the Oviedo Convention (1997).

**Informed Consent Statement:** Informed consent was obtained from all subjects and respective legal guardians involved in the study.

**Data Availability Statement:** The data presented in this study are available upon request from the corresponding author.

**Acknowledgments:** The authors would like to thank all players and respective legal guardians for participating in this study.

**Conflicts of Interest:** The authors declare no conflict of interest.

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
