# Peer review of "Associations between Age, Body Composition, Balance, and Other Physical Fitness Parameters in Youth Soccer"

_sustainability, doi:10.3390/su142013379_

Round 1

Reviewer 1 Report

The manuscript entitled “Associations between age, body composition, balance, and other physical fitness parameters in youth soccer” approaches an interesting topic related to youth development. This study's aims were twofold: (1) to analyze the balance performance of adolescent male soccer players from different age groups; and (2) to examine the relationship between players’ age, body composition, balance, and other physical fitness parameters, such as strength, and flexibility. One hundred and twelve male soccer players were assessed using bio-eletrical impedance (body composition), Jamar handgrip (muscle strength), Biodex (balance) and other field tests assessing physical fitness. Overall, this work collected information which can be relevant to discuss youth development in young athletes, despite its cross-sectional design. However, there are some issues across the manuscript which can hinder its publication in the current form, then I encourage the authors consider them. The main issues are described as per bellow.

1) The literature review reports studies the effectiveness of “balance training” (lines 65-72). However, this manuscript did not approach it. Thus, in the current form, to report these previous studies seem not linked to research question. Further, it’s considered that “…details regarding the interrelationships between balance and other physical fitness variables, such as strength and body composition, are still lacking.” That is, the authors reported a gap in the literature, but they did not present what is already known about the topic. This is a weakness in the background.

2) This work emphasizes age differences [e.g. in the title, in the introduction (lines 75-81), comparison groups, data analysis and conclusions (“Our detailed study indicates crucial differences in the balance assessment among youth soccer players according to age”)]. However, development is age-related but not age-dependent. This is a cross-sectional study and, as such, it is difficult to make comparisons among distinct ages without considering other aspects like biological and/or environmental aspects. I suggest to consider a section or paragraph with the study’s limitations, including the above mentioned issues.

3) In addition, in the background is stressed the influence of age: “…age has been proven to influence physical fitness performance, particularly in youth…” (lines 76-77); “balance strategies during gait are task-specific and vary according to age” (lines 78-79). Indeed, both physical fitness and gait patterns vary according to age. However, it is important to consider other aspects, such as those cited in the comment above (number 2). Furthermore, this background does not clarify what is already known about the relationship between age and balance performance in youth soccer players. To consider that “…this relationship is still in its infancy” is not enough for the readers to know the state of art about the topic. I strongly encourage to provide a comprehensive picture about this topic in the background.

4) Still considering the age, what are the hypotheses of this study? Did you expect an improvement or worsen of performance according to advancing age? And why? In the literature review is reported that performance improves as age increases, yet part of results (“The older group performed significantly worse in the stability indexes […] for both legs – lines 206-207) reveals the opposite.

5) It’s not clear what is the relevance of this study. What does this study add?

6) I suggest to conceptualize “physical fitness” in the Introduction.

7) In the abstract is missing some important information as the background, participants (sample size, participants per group, demography), instruments. Abbreviations should be avoid or not used in the first appearance.

Author Response

1) The literature review reports studies the effectiveness of “balance training” (lines 65-72). However, this manuscript did not approach it. Thus, in the current form, to report these previous studies seem not linked to research question. Further, it’s considered that “…details regarding the interrelationships between balance and other physical fitness variables, such as strength and body composition, are still lacking.” That is, the authors reported a gap in the literature, but they did not present what is already known about the topic. This is a weakness in the background.

Response 1: We appreciate the reviewer's feedback on the Introduction section. We have now updated the Introduction considering your valuable insights and feel that this section has improved in its quality.

2) This work emphasizes age differences [e.g. in the title, in the introduction (lines 75-81), comparison groups, data analysis and conclusions (“Our detailed study indicates crucial differences in the balance assessment among youth soccer players according to age”)]. However, development is age-related but not age-dependent. This is a cross-sectional study and, as such, it is difficult to make comparisons among distinct ages without considering other aspects like biological and/or environmental aspects. I suggest to consider a section or paragraph with the study’s limitations, including the above mentioned issues.

Response 2: We appreciate the reviewer’s important feedback. We underlined the study’s limitations as follows: “This study is based on cross-sectional data and did not assess the players’ maturity status, representing its limitations. Indeed, individual development is age-related, particularly during adolescence. Therefore, longitudinal data would be far more informative regarding youngsters’ profiles.” (line 339-341)

3) In addition, in the background is stressed the influence of age: “…age has been proven to influence physical fitness performance, particularly in youth…” (lines 76-77); “balance strategies during gait are task-specific and vary according to age” (lines 78-79). Indeed, both physical fitness and gait patterns vary according to age. However, it is important to consider other aspects, such as those cited in the comment above (number 2). Furthermore, this background does not clarify what is already known about the relationship between age and balance performance in youth soccer players. To consider that “…this relationship is still in its infancy” is not enough for the readers to know the state of art about the topic. I strongly encourage to provide a comprehensive picture about this topic in the background.

Response 3: We have updated the Introduction section based on the reviewer’s feedback.

4) Still considering the age, what are the hypotheses of this study? Did you expect an improvement or worsen of performance according to advancing age? And why? In the literature review is reported that performance improves as age increases, yet part of results (“The older group performed significantly worse in the stability indexes […] for both legs – lines 206-207) reveals the opposite.

Response 4: As suggested by the reviewer, the hypotheses of this study were added at the end of the Introduction section.

5) It’s not clear what is the relevance of this study. What does this study add?

Response 5: We thank the reviewer for the feedback. We have now reinforced the relevance of this study at the end of the Discussion section as follows: “Our results, together with the literature, which has described the strong and positive association between balance, motor skills acquisition, motor performance enhancement, and injury prevention [55-57], emphasize the need to promote training strategies to improve balance among youngsters with fewer balance abilities. This would benefit soccer game performance, particularly in actions such as dribbling, passing, and gaining positions between opponents. Due to the interrelationship between strength and balance, strength contents could be used to improve balance. Besides, body composition should be monitored, mainly to avoid the detrimental effects of BF% on balance performance.“ (line 350-358)

6) I suggest to conceptualize “physical fitness” in the Introduction. 

Response 6: As suggested, we have conceptualized “physical fitness” in the Introduction section as follows: “To the best of our knowledge, among physical fitness components, which comprise health-related (body composition, cardiorespiratory endurance, flexibility, muscular strength and endurance) and skill-related factors (balance, agility, and coordination) [20], balance is one of the less studied in youth sports.”(line 64-68)

7) In the abstract is missing some important information as the background, participants (sample size, participants per group, demography), instruments. Abbreviations should be avoid or not used in the first appearance.

Response 7: The abstract is now rewritten considering the reviewer’s feedback.

Author Response

General Comments to Author(s):

1) The authors have provided a well-designed study to address an underserved area in youth sports. However, the manuscript contains various issues that need further attention. See below for specific comments.

Response 1: We thank the reviewer for the overall positive feedback. We have addressed the specific comments and feel that the manuscript quality has improved significantly.

Specific Comments

2) Pg.2 Lines 69-73: Consider rewording for clarity.

Response 2: As suggested, we updated this text section as follows: “Although the effects of balance training to improve health and skilled-related components, such as sprinting, jumping, and sports-specific skills among young athletes is well established [23], details are still needed concerning the interrelationship between chronological age (CA) balance, body composition and strength performance in youth soccer.” (line 77-81)

3) Methods 2.4: How much recovery time occurred between jump attempts?

Response 3: 30 s of rest was given between jump attempts. This information is presented as follows: “Both protocols included four data collection trials performed 30 s apart.” (line 140-141)

4) Methods: How much recovery times was given between each exercise?

Response 4: The rest interval was nearly 5 minutes between each different protocol. This information is presented as follows: “All the assessments took place in a physical performance laboratory 5 minutes apart between protocols.” (line 101)

5) Pg. 3, Line 143: Consider rewording to remove the use of personal pronouns (their).

Response 5: The sentence was reworded as suggested.

6) Consider including potential reasons for younger players better performance for static balance and older players better performance for dynamic balance.

Response 6: We have now included more information regarding these results in the Discussion section as follows: “Several factors, among others, may influence balance ability, such as age, height, body mass, and sport participation level [44]. Previous research among 130 youth male soccer players aged between 10 and 18 years, reported similar results to the ones found in our study. Although the methods used to assess balance differed from ours, younger players significantly outperformed their peers in static balance, while older players showed better dynamic balance [45]. Indeed, the literature has mentioned that postural sway has been shown to increase with age, while timed unipedal balance tends to decrease with age [44], which is corroborated by our results.” (line 281-288)

7) Additionally, consider including further analysis on the relationship between sit-up performance and balance.

Response 7: We added more information regarding the relationship between these variables in the Discussion section as follows: “Past research described high levels of activity of trunk muscles when necessary to stabilize the trunk over a base of support [50,51], which emphasizes the role of the core in balance tasks.” (line 318-320)

8) Furthermore, consider including information on how the results of the current study could potentially be applied to performance.

Response 8: We have followed the reviewer’s advice, and introduced practical implications in the Discussion section as follows: “Our results, together with the literature, which has described the strong and positive association between balance, motor skills acquisition, motor performance enhancement, and injury prevention [55-57], emphasize the need to promote training strategies to improve balance among youngsters with fewer balance abilities. This would benefit soccer game performance, particularly in actions such as dribbling, passing, and gaining positions between opponents. Due to the interrelationship between strength and balance, strength contents could be used to improve balance. Besides, body composition should be monitored, mainly to avoid the detrimental effects of BF% on balance performance.” (line 350-358)

Reviewer 3 Report

I have no comments to the authors. Manuscript is well written. 

 In my opinion the manuscript "Associations between age, body composition, balance, and other physical fitness parameters in youth soccer" is properly planned and written. The topic of the work is new and the number of the manuscripts in this topic is still insufficient. Assessment of young football players is important in order to properly plan training, preparation for the match, and assess the risk of injury. In my opinion the manuscript is well written, introduction present the most important information which introduce reader to the main topic of the work. Part materials and methods attract attention - all the analyzes that have been carried out are described in detail. Also, results of the work was clearly presented.

The discussion of the thesis could contain newer literature, while the conclusions of the thesis should be presented in the form of points, not continuous text.

Author Response

Response 1: The authors would like to thank the reviewer for this positive feedback.

Reviewer 4 Report

I congratulate the authors, the manuscript is pretty well finished, it is a very interesting work on static and dynamic balance and their correlation with physical qualities...

Author Response

(The authors gave the same response as above.)

Reviewer 5 Report

Dear respected authors,

1. The study's main aim and content have been reflected well in the Abstract section. It is suggested to briefly mention the one-way ANOVA and the other used statistical tests in this section.

2. Some Acronyms/Abbreviations have been used in the Abstract and other sections without their complete phrases. Generally, first, an Acronym/Abbreviation should be defined, and then it should be used in the text. Check the whole text considering this issue.

3. Keywords should be selected based on the frequent usage of the phrases in the text. Considering this issue, “youth athletes” is not an appropriate keyword, as it has not been used in the text at all.

4. The aim of the study and the research gap based on the related literature has been highlighted in the Introduction section well.

5. It is suggested to make a table containing the information explained in the first paragraph of the Materials and Methods section for easy understanding the potential readers. The explanations should be referred to the table afterward.

6. The content of each sub-sections in the Materials and Methods section should be supported by references if the definition is based on standards.

7. It is recommended to define the hypothesis test, containing null and alternative hypotheses at the end of the second section.  

8. According to the results presented in Table 1, Flexibility unilateral, Flexibility bilateral, and SI ECSS are not significant. It is suggested to explain more about these results, and the potential or scientific reasons. It is highly recommended to compare the contained results to the findings of similar studies in the literature.

9. The results obtained in Tables 2 to 4 should be explained more. There are some results with no explanation about them in these tables.

10. The Conclusion section should contain briefly the used methods, and should have a separate paragraph for the limitations of the study as well as the potential future studies. 

Author Response

1.The study's main aim and content have been reflected well in the Abstract section. It is suggested to briefly mention the one-way ANOVA and the other used statistical tests in this section. Some Acronyms/Abbreviations have been used in the Abstract and other sections without their complete phrases. Generally, first, an Acronym/Abbreviation should be defined, and then it should be used in the text. Check the whole text considering this issue.

Response 1: We appreciate the reviewer's comment and have now update the Abstract section.

  1. Keywords should be selected based on the frequent usage of the phrases in the text. Considering this issue, “youth athletes” is not an appropriate keyword, as it has not been used in the text at all.

Response 2: As suggested, we have replaced “youth athletes” by a more appropriate term “youngsters”.

3.The aim of the study and the research gap based on the related literature has been highlighted in the Introduction section well. 

Response 3: We appreciate the reviewer’s positive feedback on the Introduction section. 

  1. It is suggested to make a table containing the information explained in the first paragraph of the Materials and Methods section for easy understanding the potential readers. The explanations should be referred to the table afterward.

Response 4: The data presented is this paragraph is available in Table 1. Therefore, as suggested by the reviewer, we have removed the text in the Materials and Methods section.

  1. The content of each sub-sections in the Materials and Methods section should be supported by references if the definition is based on standards.

Response 5: We have added references for each protocol mentioned in the Materials and Methods section.

  1. It is recommended to define the hypothesis test, containing null and alternative hypotheses at the end of the second section.  

Response 6: As suggested, we added the study’s hypotheses in the end of the Introduction section (line

  1. According to the results presented in Table 1, Flexibility unilateral, Flexibility bilateral, and SI ECSS are not significant. It is suggested to explain more about these results, and the potential or scientific reasons. It is highly recommended to compare the contained results to the findings of similar studies in the literature.

Response 7: As suggested, we added some information in the Discussion section regarding the flexibility results, as follows: “In contrast, no substantial differences were observed between groups concerning flexibility tests. The U16 players showed better performance levels both in the unilateral and the bilateral testing. According to past studies, flexibility tends to decrease over age due to the diminution of the range of motion [39-41], which underlines the need to consider systematic training specifically focused on developing this capacity [42].” (line 272-276)

  1. The results obtained in Tables 2 to 4 should be explained more. There are some results with no explanation about them in these tables.

Response 8: We appreciate the reviewer’s feedback on this topic. We have updated the Results section, adding more information regarding the results presented in Tables 2 to 4 in the text. In the Discussion section, we feel that we have focused on the most important findings of our study. However, the Discussion is also updated with more information concerning some variables which were not totally considered before. We hope that these updates correspond to the reviewer expectations.

  1. The Conclusion section should contain briefly the used methods, and should have a separate paragraph for the limitations of the study as well as the potential future studies.

Response 9: As suggested, we have now updated the Conclusion section as follows: “This is a cross-sectional study, and players’ maturity status was not assessed, representing its limitations. Longitudinal data and considering youngsters’ maturity status would be far more informative and should be considered in future works. Even though, our results suggest including both types of balance assessment (static and dynamic), as complementary measures while evaluating youngsters’ postural balance.” (line 371-376)

Round 2

Reviewer 1 Report

The authors kindly answered all questions and the manuscript has improved its overall quality.

Reviewer 5 Report

Dear respected authors,

The comments have been answered patiently and the text has been modified based on the raised suggestions and recommendations correctly. Therefore the manuscript is accepted but just the following issues should be rechecked before publishing the manuscript in the respected journal.

1-      In the Abstract section, the respected authors added “(-0.23 > r < -0.51).” It seems it must be corrected as “(-0.23 > r > -0.51)”.

2-      In the keywords list, the authors mentioned they added: “youth athletes” but in the revised version of the manuscript “youngsters” have been added, which is not wrong, but it is suggested to be rechecked before publication.